# Proprotein Convertase Subtilisin/Kexin 6 in Cardiovascular Biology and Disease

**DOI:** 10.3390/ijms232113429

**Published:** 2022-11-03

**Authors:** Qingyu Wu, Shenghan Chen

**Affiliations:** 1Cyrus Tang Hematology Center, Collaborative Innovation Center of Hematology, State Key Laboratory of Radiation Medicine and Prevention, Soochow University, Suzhou 215123, China; 2Vascular Function Laboratory, Human Aging Research Institute and School of Life Science, Jiangxi Key Laboratory of Human Aging, Nanchang University, Nanchang 330031, China

**Keywords:** atrial septal defect, cardiac aging, corin, endothelial lipase, hypertension, PACE4, PCSK6, vascular remodeling

## Abstract

Proprotein convertase subtilisin/kexin 6 (PCSK6) is a secreted serine protease expressed in most major organs, where it cleaves a wide range of growth factors, signaling molecules, peptide hormones, proteolytic enzymes, and adhesion proteins. Studies in *Pcsk6*-deficient mice have demonstrated the importance of Pcsk6 in embryonic development, body axis specification, ovarian function, and extracellular matrix remodeling in articular cartilage. In the cardiovascular system, PCSK6 acts as a key modulator in heart formation, lipoprotein metabolism, body fluid homeostasis, cardiac repair, and vascular remodeling. To date, dysregulated PCSK6 expression or function has been implicated in major cardiovascular diseases, including atrial septal defects, hypertension, atherosclerosis, myocardial infarction, and cardiac aging. In this review, we describe biochemical characteristics and posttranslational modifications of PCSK6. Moreover, we discuss the role of PCSK6 and related molecular mechanisms in cardiovascular biology and disease.

## 1. Introduction

Proteolytic cleavage is a key mechanism in regulating protein structure and function. The proprotein convertase subtilisin/kexin (PCSK) family consists of nine proteolytic enzymes that process a variety of proteins, including peptide hormones, growth factors, cell receptors, proteases, and adhesion molecules [1,2]. The PCSK-mediated pathways are essential for tissue homeostasis and physiological function. PCSKs are also exploited by pathogens to boost their infectivity. In COVID-19, for example, PCSK-mediated cleavage of spike protein on the surface of severe acute respiratory syndrome (SARS) coronavirus-2 particles is a crucial step for viral entry in human airways [3,4].

PCSK6, also known as paired basic amino acid cleaving enzyme 4 (PACE4) or subtilisin-like proprotein convertase 4 (SPC4), is a member of the PCSK family. PCSK6-mediated protein cleavage has been implicated in diverse biological processes, ranging from embryonic development to tissue senescence. Human genetic and animal model studies in recent years indicate that PCSK6 is an important regulator in cardiovascular biology and disease. In this review, we briefly describe the current knowledge regarding PCSK6 expression, biochemical and cellular mechanisms in zymogen activation, and protein substrate specificity. We also examine findings in *Pcsk6*-deficient mice under various conditions. Finally, we focus our discussion on the role of PCSK6 in heart development and aging, lipoprotein metabolism, blood pressure, and cardiovascular remodeling.

## 2. *PCSK6* Gene and Tissue Expression

PCSK6 was identified from human hepatoma cells based on sequence similarities to other subtilisin-like proteases [5]. The human *PCSK6* gene is located on chromosome 15 at q26.3, with 23 exons in a locus spanning ~228 kb [6]. *PCSK6* orthologues are conserved in all vertebrates from fish to primates, an indication of functional significance.

*PCSK6* mRNA expression has been detected in virtually all major organs, including the brain, heart, lung, liver, spleen, pancreas, kidney, skeletal muscle, uterus, ovary, and placenta [7]. Within the heart, high levels of *PCSK6* mRNA are expressed in atrial and ventricular cardiomyocytes [8]. Based on proteomic analysis of human tissues, PCSK6 is also expressed in cardiac fibroblasts, endothelial cells, and smooth muscle cells (SMCs) [9]. It remains unclear if PCSK6 cleaves similar sets of proteins in different cardiac cell types.

In addition to normal tissues, upregulated *PCSK6* expression has been found in many cancers, including lung cancer [10,11], breast cancer [12,13], prostate cancer [14,15], skin cancer [16,17], ovarian cancer [18,19], and thyroid cancer [20]. To date, PCSK6-mediated cleavage of metalloproteinases [17,21], signaling molecules [22,23], and apoptotic proteins [23,24] has been reported as a potential mechanism in tumor invasion and migration. In animal cancer models, PCSK6 inhibitors have been shown to inhibit tumor progression, indicating that PCSK6 is a potential cancer target [16,25,26]. In this review, our discussions focus on the role of PCSK6 in cardiovascular biology. More information on PCSK6 and cancers can be found in several comprehensive reviews [27,28,29].

## 3. Protein Domains and Post-Translational Modifications

The full-length human PCSK6 (NCBI protein accession number: NP_002561.1) consists of 969 amino acids, with a calculated mass of ~100 kDa. There are PCSK6 isoforms with shorter sequences from alternative mRNA slicing [6,14]. In cancer tissues, PCSK6 isoforms may differ from those in normal tissues [20]. Unlike furin, which has a transmembrane domain [1], PCSK6 is a secreted protein, consisting of an N-terminal signal peptide, a pro-domain, a subtilisin-like catalytic domain, a P or Homo B domain, and a C-terminal cysteine-rich domain [30] (Figure 1). Within the catalytic domain, the conserved active residues Asp, His, and Ser are at positions 205, 246, and 420, respectively.

PCSK6 is synthesized in a one-chain zymogen form. After the signal peptide is removed by signal peptidase in the endoplasmic reticulum (ER), PCSK6 undergoes a two-step autoactivation process, which resembles those in furin and PC5A activation [31,32]. The first autocleavage at RVKR_149_↓ between the pro-domain and the catalytic domain occurs in the ER [33] (Figure 2). The cleaved pro-domain is attached to the remaining fragment, thereby blocking PCSK6 catalytic activity. Moreover, the cleaved pro-domain acts as an intramolecular chaperone in protein folding and subsequent ER exiting [34]. Upon reaching the cell surface, PCSK6 binds to membrane-associated heparan sulfate proteoglycans and undergoes another autocleavage at KR_117_↓ within the pro-domain (Figure 2) [32]. This second cleavage removes the pro-domain, converting PCSK6 to a fully active enzyme [32].

The secretion and activation of PCSK6 are regulated by its own structural elements and interactions with other intracellular proteins. Within the cell, for example, association with reticulocalbin-3, a protein in the secretory pathway, promotes PCSK6 secretion and autoactivation [35]. The C-terminal cysteine-rich domain is inhibitory for PCSK6 intracellular trafficking. Deletion of a C-terminal segment or the entire domain increases PCSK6 secretion and autoactivation [36,37]. In addition to heparan sulfate proteoglycans on the cell surface [32,38], PCSK6 also binds to heparan sulfate proteoglycans in the extracellular matrix where it may activate matrix metalloproteinases [39]. In the cell-derived conditioned medium, PCSK6 also binds to heparin, which enhances PCSK6 activity [39]. The heparin-binding site on PCSK6 is in a positively charged segment, between residues 734 and 760, within the C-terminal cysteine-rich domain [39]. There is no evidence to indicate that PCSK6 binds to chondroitin sulfate [32,39]. To date, the heparan sulfate structures required for PCSK6 binding have not been determined experimentally [39]. Further studies will be important to define the biochemical and cellular mechanisms underlying PCSK6 intracellular trafficking and secretion.

In human PCSK6, there are three predicted N-glycosylation sites at N_259_, N_914_, and N_932_, respectively (Figure 1). These sites are conserved in PCSK6 homologs among mammalian species, with dogs being an exception, which have only two predicted N-glycosylation sites (corresponding to N_259_ and N_932_ in humans, respectively). In trypsin-like serine proteases, N-glycosylation facilitates protein folding, intracellular trafficking, cell surface expression, or secretion [40,41,42,43]. N-glycan-mediated interactions with calnexin, an ER chaperon, are critical for protein folding and ER exiting [41,44]. To date, the significance of N-glycosylation in PCSK6 remains unclear. Additional studies are required to verify if those predicted sites are N-glycosylated and to understand the potential role of N-glycans in PCSK6 biosynthesis and function.

## 4. Protein Substrates

Proteases break peptide bonds at preferred amino acid sequences. PCSK6 cleaves at sequences with paired basic residues, e.g., R-X-K/R-R, where X can be any amino acid, and K/R-R or non-paired basic residues, e.g., R-X-X-R [45,46]. These recognition sequences, however, are not unique for PCSK6, as many PCSKs exhibit similar substrate specificities in biochemical and cellular experiments [1]. Under physiological conditions, the function of individual PCSKs also depends on their spatial and temporal expression and regulatory mechanisms in specific tissue environments.

To date, numerous proteins have been reported as potential PCSK6 substrates. A partial list of the candidate substrates includes growth factors [47,48], cell surface receptors [49,50], blood clotting factors [51,52], matrix metalloproteinases (MMPs) [17,21], serine proteases [53,54], lysyl oxidase-like 2 [55], group X secretory phospholipase A2 [56], growth differentiation factor 15 [57], Nodal [58], bone morphogenetic protein-4 (BMP-4) [58,59], neural and hepatic peptides [60,61], bacterial toxins [45,62], and viral proteins [63,64]. These findings suggest a potential role of PCSK6 in diverse pathophysiological processes, ranging from embryonic development to tissue homeostasis to bacterial and viral infection. It is noted that the majority of the reported PCSK6 substrates are based on in vitro experiments. Many of the substrates are also cleaved by other PCSKs in similar experimental settings. It remains to be determined if the observed overlapping activities reflect the functional redundancy among PCSKs in vivo.

## 5. Findings from *Pcsk6* Knockout (KO) Mice

### 5.1. Embryonic Development

In mice, *Pcsk6* deficiency leads to lethality in ~25% embryos, which exhibit severe defects in heart formation, craniofacial patterning, and left–right axis specification by gestational day 15.5 [65]. The phenotype is primarily due to impaired processing and signaling of transforming growth factor beta (TGFβ) superfamily members, including Nodal, Bmp4, and Lefty, that are crucial in body axis specification during embryonic development [65,66,67]. These findings are consistent with the observed PCSK6 activity in processing TGFβ-like growth factors in vitro [59,65].

The incomplete penetrance of the embryonic lethality in *Pcsk6* KO mice suggests that the Pcsk6 function in embryogenesis may be partially compensated by other PCSKs, particularly furin that exhibits similar Nodal and BMP4 processing activity in vitro and in vivo [58,59]. Moreover, both PCSK6 and furin are shown to bind to Cripto, a glycosylphosphatidylinositol (GPI)-anchored proteoglycan that localizes Nodal processing on the surface of embryonic cells [68]. It remains unclear how PCSK6 and furin activities are regulated, temporally and spatially, in embryonic tissues to achieve the overlapping but not identical function.

### 5.2. Age-Dependent Decline in Ovarian Function

The PCSK6 function in processing TGFβ-like growth factors appears dispensable for postnatal survival. The remaining three quarters of *Pcsk6* KO mice that are born grow normally [65,69]. Both male and female *Pcsk6* KO mice are fertile [65]. No major defects in liver and kidney function, blood coagulation, immune response, and tooth formation have been reported in these mice.

Studies in aged *Pcsk6* KO mice, however, indicate a key role of Pcsk6 in preserving ovarian structure and function [69]. Female *Pcsk6* KO mice at or older than nine months exhibit decreased estrus cyclicity, prolonged whelping intervals, increased serum follicle-stimulating hormone levels, and ovarian atrophy with less or no follicles [69]. Given the findings of *PCSK6* expression in human and mouse ovarian granulosa cells and oocytes [70,71,72], the reproductive senescence in female *Pcsk6* KO mice is likely related to impaired processing of Pcsk6 substrate(s) in the ovary. To date, such substrate(s) have yet to be defined. Among TGFβ superfamily members, oocyte-derived growth differentiation factor 9 (Gdf9) and Bmp15 are essential for folliculogenesis in mice [73]. In human genetic studies, *GDF9* and *BMP15* variants are found in women with premature ovarian failure and amenorrhea [74,75,76], suggesting that GDF9 and BMP15 could be potential PCSK6 substrates in the ovary.

### 5.3. Aggrecan Degradation in Articular Cartilage

Analysis of *Pcsk6* KO mice substantiates a role of PCSK6 in aggrecan degradation and osteoarthritis [77]. Aggrecan is a proteoglycan with abundant chondroitin and keratan sulfate chains [78,79]. It aggregates on hyaluronan filaments in the extracellular matrix of articular cartilage, protecting the joint from the impact of compressive forces [78,79]. Increased aggrecan degradation is a key factor in the pathogenesis of osteoarthritis [80]. Members of the ADAMTS (a disintegrin and metalloproteinase with thrombospondin motifs) family, mainly ADAMTS-4 and ADAMTS-5, are major aggrecan-degrading enzymes in osteoarthritis [78,80].

In human osteoarthritic cartilage, PCSK6 has been identified as a key activator of ADAMTS-4 and ADAMTS-5 [81]. In mice, *Pcsk6* deficiency alleviates pain in osteoarthritis models [77]. A *PCSK6* variant allele is also associated with similar pain protection in patients with osteoarthritic knees [77]. Moreover, increased PCSK6 expression is found in synovial tissues in patients with rheumatoid arthritis [82]. PCSK6 inhibition reduces inflammatory responses in rat and human synoviocytes associated with rheumatoid arthritis [82,83,84]. These results highlight a crucial role for PCSK6 in cartilage remodeling and destruction under pathological conditions.

## 6. PCSK6 and Cardiovascular Pathophysiology

### 6.1. Atrial Septal Defects

Atrial septal defects are a common form of congenital heart disease [85]. As discussed earlier, *Pcsk6* KO mice exhibit a plethora of cardiac abnormalities, including common atrium, double-outlet right ventricle, ventricular septal defects, persistent truncus arteriosus, and dextrocardia [65]. Common atrium is a severe type of atrial septal defect, in which the entire atrial septum is missing. In mice, Nodal and Bmp4 are downstream effectors of Pcsk6 in heart formation [65,66]. However, the upstream molecular network that regulates PCSK6 expression in developing hearts is not well defined.

TBX5, NKX2-5, and GATA4 are major transcription factors in heart development. Deleterious *TBX5*, *NKX2-5*, and *GATA4* mutations are found in individuals with congenital heart disease, including atrial septal defects [86,87]. Protein odd-skipped-related 1 (OSR1) is a zinc-finger transcription factor, essential for cardiac progenitor growth and atrial septum formation [88]. In mouse embryonic hearts, *Tbx5* is an immediate *Osr1* upstream gene in the second heart field for atrial septation [89]. *Osr1* deletion results in common atrium and embryonic death between E11.5 and E12.5 days [90]. The similar phenotype of atrial septal defects in *Tbx5*-, *Osr1*-, and *Pcsk6*-deficient mice suggests a possible TBX5-OSR1-PCSK6 pathway in promoting TGFβ-like growth factor signaling in atrial septum formation.

In agreement with this hypothesis, gene profiling in *Tbx5*- and *Osr1*-deficient embryos identifies *Pcsk6* as one of the major genes in atrial septation regulated by *Tbx5* and *Osr1* [91]. Human genome-wide linkage analysis also suggests a connection between *PCSK6* and congenital heart disease [92]. Moreover, a *PCSK6* variant is found in a Spanish family with atrial septal defects and interatrial septal aneurysm [91]. These findings indicate that TBX5 and OSR1 are possible regulators in PCSK6 expression during heart development, providing new insights into the genetic mechanisms underlying atrial septal defects.

### 6.2. Endothelial Lipase Inactivation

Endothelial lipase, a member of the triglyceride lipase family, is a secreted protein, consisting of a signal peptide, a 40 kDa N-terminal catalytic domain, and a 28 kDa C-terminal domain [93]. Upon secretion, endothelial lipase binds to heparan sulfate proteoglycans on the cell surface, where it hydrolyzes triglycerides and phospholipids in plasma lipoproteins [94]. In addition to endothelial cells, endothelial lipase is expressed in hepatocytes, macrophages, and SMCs. Endothelial lipase modifies high-density lipoprotein (HDL) structure and metabolism, as indicated by high levels of plasma HDL cholesterol, bigger HDL particle sizes, and slower HDL clearance in endothelial lipase-deficient mice [95,96]. Conversely, plasma HDL cholesterol levels are decreased in endothelial lipase overexpressing mice [96,97]. In humans, deleterious variants in the *LIPG* gene, encoding endothelial lipase, are associated with increased plasma HDL cholesterol levels [98,99,100].

In membrane-bound serine proteases, ectodomain cleavage is a common mechanism in limiting protease activity on the cell surface [101,102]. A comparable mechanism exists in endothelial lipase inactivation on the cell surface. In the conditioned medium from human endothelial cells, a 40 kDa endothelial lipase fragment was detected [46,97]. A similar fragment was also found from human liver HepG2 cell culture [103]. As revealed by biochemical analyses, the fragment is derived by cleavage at a specific site, RNKR↓, which reduces the endothelial lipase activity [46,103]. Furin, PC5A, and PCSK6 are likely responsible for the cleavage [46,103]. Interestingly, lipoprotein lipase, another member of the triglyceride lipase family, contains an analogous site, RAKR↓, which is also cleaved by furin, PC5A, and PCSK6 in similar experiments [46]. These data indicate that PCSK-mediated endothelial lipase inactivation is a cellular mechanism in regulating lipoprotein metabolism.

Consistently, hepatic overexpression of pro-furin, an inhibitor of furin, PCSK5, and PCSK6 reduce endothelial lipase inactivation and lower plasma HDL cholesterol in mice [104]. In *Lipg* KO mice, such an effect of pro-furin on plasma HDL cholesterol is not observed, supporting a PCSK-endothelial lipase-dependent mechanism in HDL metabolism. Studies in those mice also validate a role of PCSK-mediated activation of an endogenous endothelial lipase inhibitor, angiopoietin-like 3 [104,105]. Further analyses in hepatocyte-specific *Furin* and *Pcsk5* conditional KO and *Pcsk6* global KO mice show that hepatic furin is primarily responsible for cleavage of endothelial lipase and angiopoietin-like 3 in vivo [106]. However, plasma levels of HDL cholesterol are only slightly reduced in the hepatocyte *Furin* conditional KO mice or not changed in the hepatocyte *Pcsk5* conditional and *Pcsk6* global KO mice, compared to that in WT mice [106]. These findings suggest functional redundancy among PCSKs in endothelial lipase inactivation, at least in mice.

### 6.3. Corin Activation and Hypertension

Atrial and B-type natriuretic peptides (ANP and BNP, respectively) are hormones in the natriuretic peptide system that preserves body fluid balance and cardiovascular homeostasis [107]. Genetic studies in mice and humans establish ANP as a key factor in blood pressure regulation [108,109,110]. Upon binding to its receptor, natriuretic peptide receptor A (also called guanylate cyclase A), ANP enhances renal salt excretion and relaxes blood vessels, thereby lowering blood volume and pressure. Variants in the *NPPA* gene, encoding ANP, are associated with increased risks of cardiovascular disease, such as hypertension, stroke, and heart disease [111,112].

Corin is a membrane-bound protease, highly expressed in the heart [113], where it converts pro-ANP to ANP [114,115,116]. Like most proteases, corin is produced in a pro-form, which is activated at a specific site, RMNKR↓ [117]. The cleavage sequence with paired basic residues indicates that corin is likely activated by one of the PCSKs. Indeed, PCSK6 has been identified as the corin activator [53]. Both PCSK6 and corin are expressed in cardiomyocytes, where PCSK6 activates corin on the cell surface [53,118]. In cultured murine cardiomyocytes, blocking *Pcsk6* expression prevents corin activation [53]. In *Pcsk6*-deficient mice, corin activation and pro-ANP processing in the heart are eliminated [53]. Like *Corin* KO mice, *Pcsk6*-deficient mice develop salt-sensitive hypertension [53,119], indicating that PCSK6 is the corin activator in vivo and that this function cannot be substituted by other PCSKs.

In line with these findings, genetic studies support a role of PCSK6 in corin activation and cardiovascular function in humans. For example, several *CORIN* variants identified in hypertensive patients are defective in PCSK6-mediated activation [53,120]. *PCSK6* variants are associated with hypertension [53,120] and coronary artery stenosis [121]. Studies in humans and rat models also indicate an important PCSK6-corin-ANP pathway in regulating renal aquaporin 2 and β-epithelial sodium channel expression in response to a high-salt diet [122], consistent with salt-sensitive hypertension in *Pcsk6*, *Corin*, and *Nppa* KO mice [53,109,119,123]. Moreover, reduced cardiac and renal PCSK6 and corin expression correlates with worsening cardiac and renal function in a rat heart failure model [124]. These data highlight the importance of PCSK6 in corin activation and body fluid-electrolyte homeostasis.

### 6.4. Vascular Remodeling in Atherosclerosis

Atherosclerosis is a major vascular disease, characterized by the formation of atherosclerotic plaques in the intima of medium- to large-sized arteries [125]. Depending on disease stages, the plaque usually contains lipid-packed macrophages, also called foam cells, and SMCs that are surrounded by accumulated extracellular matrix proteins and proteoglycans [125]. As the disease progresses, the macrophages and SMCs undergo apoptosis, creating a highly thrombotic necrotic core that is prone to rupture, thereby causing thrombosis formation [125]. To date, several lines of evidence point to a potential role of PCSK6 in regulating SMC migration, vascular remodeling, and atherosclerotic plaque formation.

In patients with aortic dissections in a Korean population, for example, genomic alternations are found in a locus where the *PCSK6* gene is located [126]. Genome-wide expression analysis indicates elevated *PCSK6* expression in atherosclerotic plaques [127,128]. In cultured human monocytes and endothelial cells, PCSK6 expression and activity are increased by pro-atherogenic lipid oxidation products [129]. In *Pcsk6* KO mice, compromised vascular remodeling, as indicated by enlarged systolic and diastolic circumferences and reduced contractile SMC markers, is observed in carotid arteries exposed to increased blood flow [130]. Increased PCSK6 expression is also detected in smooth muscle α-actin (SMA) (an SMC marker) -positive cells in unstable carotid plaques, where inflammation and extracellular matrix degradation are active [131]. Moreover, PCSK6 expression in cultured human carotid SMCs is increased by proinflammatory factors, such as tumor necrosis factor and interferon-γ [131]. These findings suggest a connection between PCSK6 and SMC-derived cells in the vessel wall where inflammation and pathological remodeling occur.

Consistently, a recent human study links a *PCSK6* variant with SMA-positive cell numbers in carotid stenosis lesions and artery wall thickness [132]. In human and rodent carotid arteries, increased PCSK6 expression correlates with SMC activation, intimal hyperplasia, and MMP2/MMP14 activation [132]. Conversely, decreased intimal hyperplasia and MMP14 activation are found in *Pcsk6* KO mice with carotid artery ligation [132]. Moreover, aortic SMCs from *Pcsk6* KO mice exhibit poor proliferation and migration induced by platelet-derived growth factor BB (PGDFBB), whereas in human SMCs overexpressing PCSK6, PDGFBB-stimulated cell proliferation and migration are increased [132]. PCSK6 is known to activate MMPs in cancers [133]. The latest findings suggest that PCSK6-mediated MMP activation may be important in SMC phenotypic changes and pathological vascular remodeling in atherosclerosis.

### 6.5. Cardiac Repair after Myocardial Infarction (MI)

MI triggers a series of cellular events, including cell death, inflammatory cell infiltration, and gradual wound healing with myofibroblast proliferation and ultimate scar formation [134]. Tissue remodeling depends on the interplay among various cell types, including immune cells, cardiomyocytes, fibroblasts, and vascular cells. Both autocrine and paracrine mechanisms are involved in cell–cell interactions in infarcted hearts [134].

Many serine proteases have been implicated in cardiac structure and function [135]. In a recent study, PCSK6 was identified as one of the highly secreted proteins from hypoxic cardiomyocytes [136]. The finding is confirmed in mouse hearts undergoing coronary artery ligation [136]. The PCSK6 expression and secretion in hypoxic cardiomyocytes promote TGFβ secretion from the same cells and subsequent SMAD (small and mothers against decapentapletic) signaling in cardiac fibroblasts [136]. Moreover, high levels of collagen I production and fibrosis-related gene expression (e.g., *Col1a1*, *Col3a1*, and *Mmp14*) are observed in cardiac fibroblasts treated with the PCSK6-containing conditioned medium from hypoxic cardiomyocytes [136]. These findings suggest that upregulated PCSK6 in ischemic cardiomyocytes activates TGFβ, which, in turn, binds to its receptor on cardiac fibroblasts, thereby enhancing downstream SMAD signaling to promote collagen production and cardiac fibrosis [136].

Increased fibrosis is a hallmark of poor cardiac remodeling, which impairs cardiac function. Consistently, PCSK6 overexpression in cardiomyocytes increases cardiac hypertrophy and fibrosis and decreases cardiac function in a mouse MI model [136]. Moreover, increased serum PCSK6 levels are observed in patients with acute MI, which peaks on day 3 post incidence [136]. Previously, increased ventricular, but not atrial, *Pcsk6* expression was noticed post MI in a rat model [137]. These data support a role of PCSK6 in a paracrine mechanism, underlying cardiac remodeling after MI. In another study [138], serum PCSK6 levels were associated with cardiovascular events in a subset of patients undergoing coronary angiography. Further studies will be important to evaluate if serum PCSK6 can be used as a biomarker to predict cardiac remodeling and function in patients with heart disease.

### 6.6. Cardiac Senescence

In aging hearts, altered protein expression and signaling often lead to deteriorating cardiac structure and function. In addition to apoptosis, senescence is a common feature in aging cardiomyocytes, as indicated by DNA damage, dysregulated gene expression, increased oxidative stress, mitochondrial dysfunction, and poor contractility [139,140]. Natriuretic peptide-mediated signaling is critical in cardiomyocyte homeostasis [111]. In humans, variants in the *NPPA* gene are associated with impaired cardiovascular responsiveness in the elderly [141]. In rodents, decreased ANP secretion is found in aging hearts and senescent cardiomyocytes in culture [142,143].

PCSK6 is necessary for corin activation and ANP generation in the heart [53]. A recent study indicates that PCSK6 deficiency may contribute to senescence in cardiomyocytes [144]. In aged mouse hearts and senescent cardiomyocytes, Pcsk6 expression is reduced. Moreover, *Pcsk6* downregulation causes senescence in cultured cardiomyocytes, as indicated by increased advanced glycation end products, oxidative stress, and apoptosis [144]. Conversely, *Pcsk6* overexpression prevents senescence and dysfunction in cultured cardiomyocytes under similar experimental conditions [144].

The function of PCSK6 in cardiomyocyte senescence appears mediated, at least in part, by pathways related to ER stress. In aging mouse hearts and *Pcsk6* knockdown cardiomyocytes, high levels of DNA-damage inducible transcript 3 (Ddit3) are observed [144]. Ddit3, also called C/EBP homologous protein, is a pro-apoptotic transcription factor inducted by ER stress [145]. In cardiomyocytes subjected to ER stress, Ddit3 expression is suppressed by PCSK6 expression [144], suggesting that PCSK6 may regulate cardiomyocyte senescence by reducing ER stress via a DDIT3-related mechanism. Consistent with these findings, increased ER stress is reported in human prostate cancer cells, in which the *PCSK6* gene is downregulated [24]. In a mouse model of heart failure, *Ddit3* deletion prevents ER-stress-induced cell death and cardiac dysfunction [146]. As discussed earlier, premature ovarian senescence is observed in *Pcsk6* KO mice [69]. It will be important to examine if similar premature aging exists in other major organs in *Pcsk6* KO mice.

## 7. Conclusions

PCSK6 is a multifunctional protease that acts in diverse tissues to modulate many pathophysiological processes, ranging from embryonic development to organ aging. Despite the apparent overlapping substrate specificity among PCSKs in biochemical and cellular studies, PCSK6 exhibits distinct physiological functions that are not fully compensated by other PCSKs, as indicated by findings in *Pcsk6*-deficient mice. The unique function of PCSK6 is likely achieved by mechanisms that regulate PCSK6 activity in specific cellular environments during various stages of life. To date, such regulatory mechanisms have yet to be fully understood.

PCSKs are important in the cardiovascular system. PCSK9, for example, is a key regulator in LDL receptor expression and lipid metabolism [147]. Currently, PCSK9 inhibitors are used to treat patients with familial hypercholesterolemia [147]. By acting on various growth factors and proteases, PCSK6 too plays a key role in cardiovascular development and homeostasis (Figure 3). To date, PCSK6 has been implicated in major cardiovascular diseases, such as atrial septal defects, hypertension, atherosclerosis, MI, and cardiac aging. More investigations are anticipated to determine if modulating PCSK6 expression and/or activity is a valid therapeutic strategy for cardiovascular disease.

## Figures and Tables

**Figure 1 ijms-23-13429-f001:**
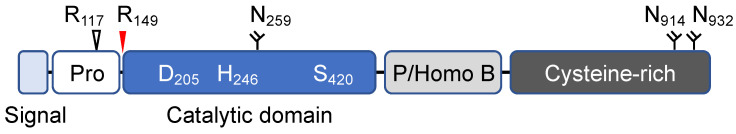
Schematic presentation of human PCSK6 domains. Human PCSK6 consists of a signal peptide (Signal), a pro-domain, a subtilisin-like catalytic domain, a P or Homo B domain, and a C-terminal cysteine-rich domain. In the catalytic domain, the conserved active sites, Asp (D), His (H), and Ser (S), are at positions 205, 246, and 420, respectively. Three predicted N-glycosylation sites (Y shaped symbols) are at positions 259, 914, and 932, respectively. Two autoactivation cleavage sites are at Arg (R) 117 (open arrowhead) (second cleavage) and R149 (red arrowhead) (first cleavage).

**Figure 2 ijms-23-13429-f002:**
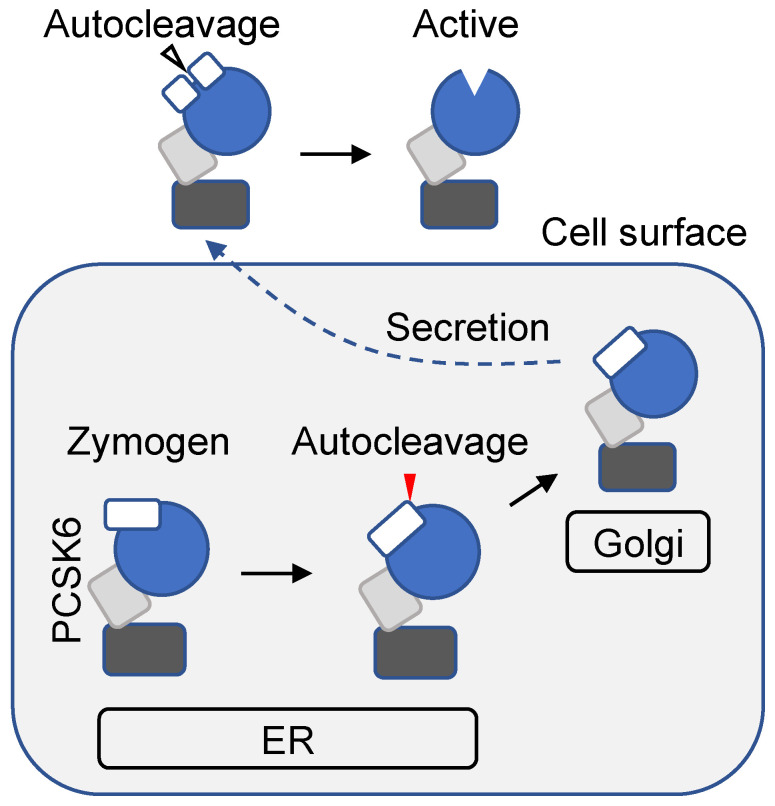
PCSK6 intracellular trafficking and autoactivation. PCSK6 is synthesized as a one-chain zymogen. The signal peptide is removed by signal peptidase in the ER (not shown). The pro-domain (white), the catalytic domain (blue), the P/Homo B domain (gray), and the C-terminal cysteine-rich domain (black) are shown. Within the ER, first autocleavage occurs between the pro-domain and the catalytic domain (red arrowhead). The cleaved pro-domain remains attached, acts as an inhibitor of PCSK6, and facilitates PCSK6 folding and ER exiting. Upon secretion, a second autocleavage occurs within the pro-domain (open arrowhead), which disassociates the pro-domain, converting PCSK6 to an active enzyme.

**Figure 3 ijms-23-13429-f003:**
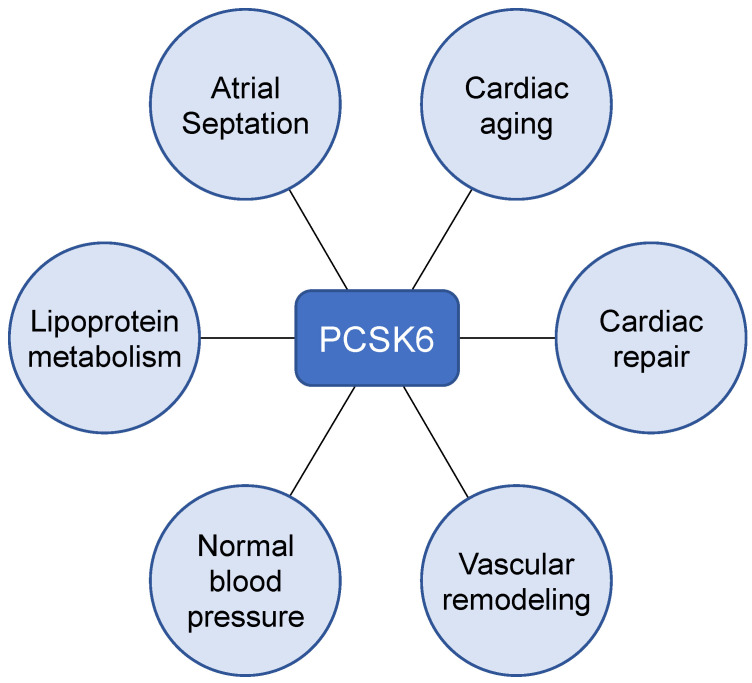
Role of PCSK6 in cardiovascular development and pathophysiology. PCSK6 plays a key role in atrial septum formation during embryogenesis. In adult stages, PCSK6 regulates lipoprotein metabolism and blood pressure by inactivating endothelial lipase and activating corin, respectively. PCSK6 participates in vascular remodeling and cardiac repair under pathological conditions such as atherosclerosis and acute MI. In aged hearts, PCKS6 prevents senescence in cardiomyocytes by reducing ER stress.

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
