# Peer review of "Proprotein Convertase Subtilisin/Kexin 6 in Cardiovascular Biology and Disease"

_ijms, 2022, doi:10.3390/ijms232113429_

Round 1

Reviewer 1 Report

Intersting and important review on important protein family

Reviewer 2 Report

This is a scientifically insightful and concise review article focusing on PCSK6.  There are some minor comments and questions below.

1.       Please elaborate a bit more on the binding of PSCK6 heparan sulfate proteoglycans.  What type heparan sulfate proteoglycans are involved, membrane-bound or secreted ones?  What would be the consequence of the binding, for the second step of the zymogen activation or simply being deposited in the ECM, or something else? 

2.       Is there any tangible evidence for which of the three putative N-glycosylation sites occupied by N-glycans?

Reviewer 3 Report

The reviewer is well-organized and the current format is of high quality. 

One weakness is that the literature is kind of out-of-date. This is largely due to a focus on cardiovascular systems and lipid metabolism, while more recent publications are in the field of cancer studies. To strengthen the review, the authors may consider adding an additional part regarding Pcsk6 in cancer studies.
